# Application of Path Analysis and Remote Sensing to Assess the Interrelationships between Meteorological Variables and Vegetation Indices in the State of Espírito Santo, Southeastern Brazil

Adriano Posse Senhorelo [1], Elias Fernandes de Sousa [2], Alexandre Rosa dos Santos [3], Jéferson Luiz Ferrari [1], João Batista Esteves Peluzio [1], Rita de Cássia Freire Carvalho [3], Kaíse Barbosa de Souza [3] and Taís Rizzo Moreira[3,*]

1   Federal Institute of Espírito Santo (IFES), Campus Alegre, BR 482, km 7, Rive, Alegre 29500-000, Brazil; apsenhorelo@ifes.edu.br (A.P.S.); ferrarijl@ifes.edu.br (J.L.F.); jbpeluzio@ifes.edu.br (J.B.E.P.)
2   Postgraduate Program in Plant Production, State University of North Fluminense Darcy Ribeiro (UENF), Av. Alberto Lamego, 2000, Campos dos Goytacazes 28013-602, Brazil; sousa.elias.fernandes@gmail.com
3   Center of Agricultural Sciences, Federal University of Espírito Santo (UFES), Alto Universitário, s/n, Alegre 29500-000, Brazil; alexandre.santos@pq.cnpq.br (A.R.d.S.); freirecarvalhor@gmail.com (R.d.C.F.C.); kaisesouza172@yahoo.com.br (K.B.d.S.)
*   Correspondence: taisr.moreira@hotmail.com

**Abstract:** Utilizing path analysis, we examined the interconnectedness among six meteorological variables. Among these, three pertain to energy conditions—air temperature, net solar radiation, and reference evapotranspiration (ET0)—while the others are associated with hydrological conditions: precipitation, relative humidity, and water deficiency. These variables were assessed across five distinct temporal delay levels to understand their influences on the normalized difference vegetation Index (NDVI) and enhanced vegetation index (EVI) within grassland areas situated in the state of Espírito Santo, southeastern Brazil. The images underwent processing using analytical algorithms and a geographic information system (GIS). The direct and indirect impacts of these variables on the NDVI and EVI exhibited remarkable similarity across varying temporal delays and geographic regions. Meteorological variables explained over 50% of the observed variation in both indices, occasionally even reaching levels of 70%. Temperature and relative humidity primarily exerted direct effects on the indices. Conversely, precipitation exhibited indirect effects on the indices, often in conjunction with other hydrological variables. ET0 demonstrated a direct effect on the vegetation indices, particularly after a delay of 32 days. Solar radiation and water deficiency displayed direct effects up to the 32-day mark, implying that vegetation responds more promptly to these variables. The proposed methodology enabled a consistent and stable assessment of the direct and indirect effects of meteorological variables on vegetation indices.

**Keywords:** enhanced vegetation index; geographic information system; geotechnologies; meteorological variables; normalized difference vegetation index; path analysis



## 1. Introduction

Climate stands as the primary driver behind the distribution of various vegetation types within distinct terrestrial ecosystems, controlling their spatial and temporal transformations at local and global scales [1–3]. Consequently, vegetation assumes a pivotal role as an indicator, given that any alterations in its quality and quantity reverberate across its environmental context [4–6]. This underscores the worldwide significance of comprehending how vegetation responds to meteorological variables, an imperative subject of research [7–12]. When a specific meteorological variable significantly impacts the state of vegetation within an ecosystem, its role gains even greater prominence, particularly within the context of climate change. Among the variables explored in the realm of plant

dynamics and growth, air temperature and precipitation emerge as focal points in the discussion [13–18]. Nonetheless, alongside these aspects, other variables wield influence over vegetation and must not be disregarded, including net solar radiation, reference evapotranspiration, and water deficiency [19,20].

Examining both the direct and indirect effects of meteorological variables on vegetation development holds paramount importance in comprehending the intricate interplay between climate change and shifts in vegetation dynamics. The integration of geotechnologies into the study of vegetation dynamics has expedited the process, rendering it economically viable and less labor-intensive. Furthermore, remote sensing has emerged as a highly effective tool across a multitude of research domains. It boasts frequent data acquisition, serves as an accessible and standardized information source across diverse spectral regions, and furnishes a global perspective on phenomena [21–25]. Spectral models like the enhanced vegetation index (EVI) and the normalized difference vegetation index (NDVI) have demonstrated their efficacy in various studies [26–34]. These images have enabled the identification of substantial changes in vegetation, given that variations in vegetation indices exhibit significant correlations with green biomass content and plant water stress [35,36].

Relying on these crucial insights, given the observation of climate shifts affecting millions of individuals and giving rise to global economic, environmental, and social repercussions, particularly within the agricultural sector, it becomes imperative to identify the meteorological variables that play a more active role in these transformations and gain a deeper understanding of their impact on vegetation dynamics. In the context of such scientific inquiries, it often becomes essential to ascertain the presence or absence of interactions among variables. This necessitates the development of correlation analyses to evaluate both the strength and direction of the relationship between two variables. However, while correlation analysis constitutes an important stage, it falls short of establishing a cause-and-effect relationship and can be misinterpreted. Elevated correlation values might arise from indirect effects exerted by other variables. In such circumstances, recourse to alternative methodologies, like path analysis, offers a means to apprehend the genuine connections between variables [37]. Path analysis entails disentangling correlations into direct and indirect effects originating from explanatory variables onto the fundamental variable. The quantification of these effects is achieved through multiple regression analysis. This dissection of correlations hinges on the predefined set of variables under study, their individual significance, and the potential interrelations depicted in a path diagram [38].

Scholars have established the versatility of path analysis across various domains, as follows. (1) Medical Studies: path analysis has been employed in dietary inflammatory index assessments to gauge its connection with maternal factors and excessive body weight in Brazilian children during complementary feeding [39]. (2) Psychological Investigations: within psychological studies, path analysis frequently evaluates caregiving burden, family functioning, and psychological well-being among older caregivers of elderly adults [40]. Additionally, there is a pronounced interest in understanding the associations between addiction, self-esteem, fear of loss, individual daily time allocation, and problematic social media use [41]. (3) Chemical and Physical Research: the methodology's significance extends to chemical and physical analyses, as seen in a study revealing the browning of ready-to-eat crayfish tails during thermal treatment and storage [42]. Furthermore, it contributes to a comprehensive understanding of adaptive thermal comfort in dynamic environments [43]. (4) Animal Husbandry Studies: in the realm of animal husbandry, path analysis proves beneficial for exploring direct and indirect effects within the context of lactation number, calving season, dry period, service period, and insemination number in Holstein cows [44]. (5) Agricultural Studies: in agricultural research, path analysis plays a pivotal role in establishing causal relationships between vegetation indices and soybean grain yield based on in situ observations [45]. (6) Genetic Improvement Research: notably, the methodology finds frequent application in genetic improvement studies. Authors have employed it in crop breeding to facilitate the formulation of suitable selection procedures [46–52].

The quantification of both the direct effect of a meteorological variable on vegetation development and its indirect effect through other meteorological factors has been infrequently undertaken. Another significant aspect is that the few studies employing path analysis to assess the interrelationships between meteorological variables and their effects on vegetation indices primarily focus solely on the effects of temperature and precipitation on the NDVI [53–55]. Therefore, as a distinctive aspect of this study, we aim to broaden the scope by incorporating a greater number of meteorological variables into the analysis process. Additionally, we utilize both the NDVI and EVI. Moreover, we take into account the temporal delay in vegetation response concerning the action of meteorological variables.

Thus, our main purpose is to utilize path analysis for the exploration of causal relationships among the subsequent meteorological variables: three variables linked to energy conditions (air temperature, net solar radiation, and reference evapotranspiration), and three variables associated with hydrological conditions (precipitation, relative humidity, and water deficiency). This investigation encompasses five distinct temporal delay levels (16, 32, 48, 64, and 80 days) in relation to vegetation's response to meteorological factors. We also aim to discern their direct and indirect effects on the NDVI and EVI within grassland areas.

## 2. Materials and Methods

### 2.1. Study Area

The study area covered the southeastern region of Brazil, specifically the state of Espírito Santo, located between the states of Bahia and Rio de Janeiro, and extending into Minas Gerais. This area also includes the Atlantic Ocean, bounded by the following coordinates: parallels 17.9° S and 21.3° S and meridians 39.6° W and 41.8° W (Figure 1a).

According to Köppen's classification, an Aw climate (tropical zone, with dry winter) predominates in the largest portion of Espírito Santo (53.69%). The study area also encompasses several other climate zones, including Cfa, which corresponds to an oceanic climate without a dry season (14.92%), Am, indicative of a tropical zone with a monsoon period (13.96%), Cfb, characterized by an oceanic climate without a dry season but with temperate summers (10.47%), Cwb, denoting a humid temperate climate with dry winters and temperate summers (3.36%), Af, representing a humid tropical climate (2.76%), Cwa, indicating a humid temperate climate with dry winters and hot summers (0.83%), and Cwc, classifying a humid temperate climate with dry winters and short, cool summers (0.02%) [56].

### 2.2. Satellite Image Acquisition and Processing

A methodological flowchart outlining the essential steps for acquiring and preprocessing NDVI, quality, and pixel reliability images is presented in Figure 1b, complemented by a summary of the developed model for automating and documenting data management processes (Figure 1c) [57].

We specifically selected NDVI and EVI images with a spatial resolution of 250 m and a temporal resolution of 16 days. These images were accessible at no cost through the website https://search.earthdata.nasa.gov. Version V06 of the datasets, corresponding to the h14v10 and h14v11 tiles, was utilized. Each tile encompasses an area of 1200 × 1200 km (Figure 1a) [28]. The imagery for the span of a decade, from 2008 to 2017, was chosen (Table 1). A total of 230 images for each layer underwent processing and served as data sources. All geoprocessing procedures and image editing tasks were conducted using ArcGIS 10.3 [58].

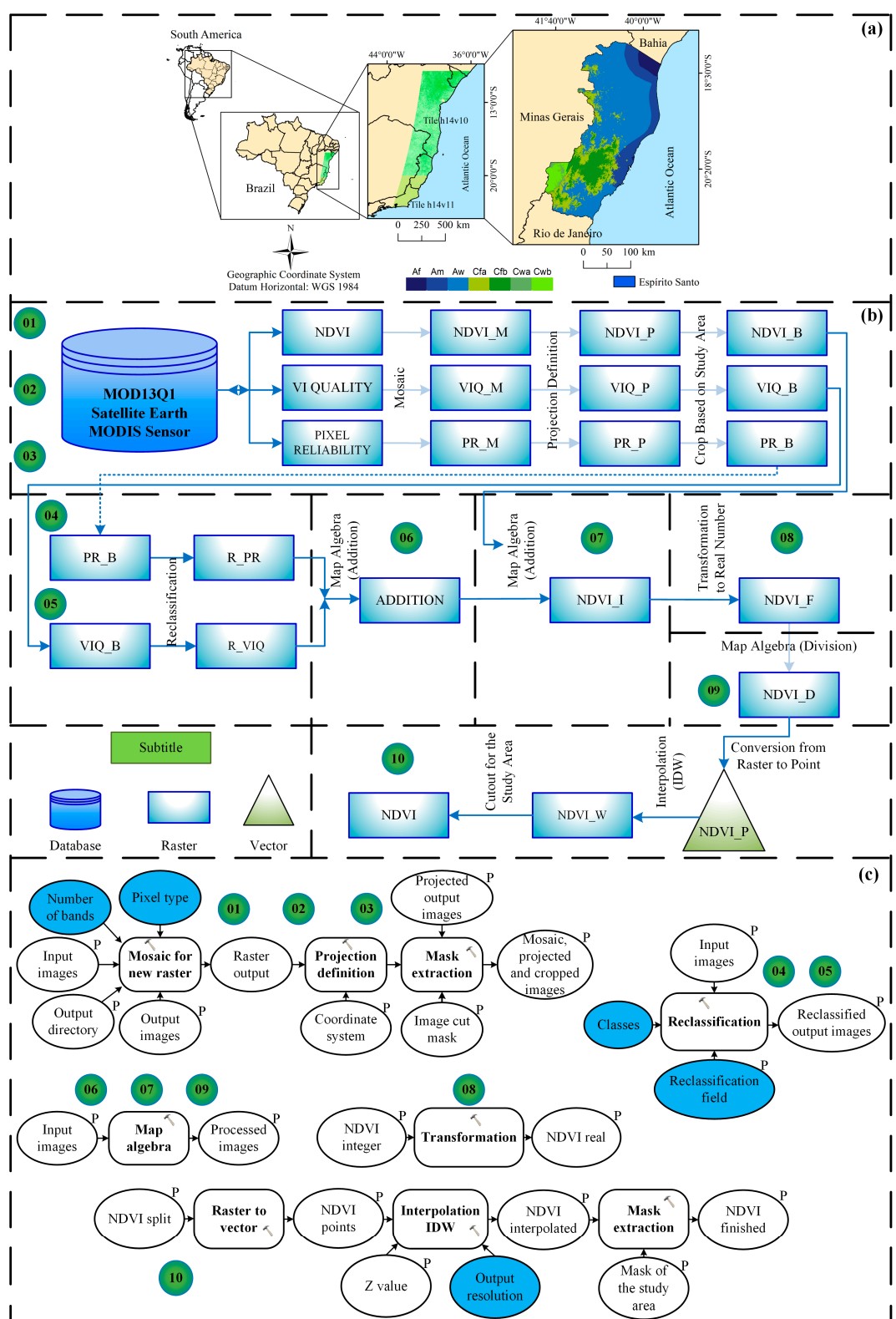

**Figure 1.** Location of the state of Espírito Santo within the established climate domains, followed by images corresponding to h14v10 and h14v11 tiles (**a**); methodological flowchart describing the preprocessing of NDVI images (**b**); and steps of the developed model (**c**).

**Table 1.** Dates of MODIS images.

| Start Date (Julian Day) | Image Years |
|---|---|
| 01/01 (001), 01/17 (017), 02/02 (033), 02/18 (049), * 03/06 (065), 03/22 (081), 04/07 (097), 04/23 (113), 05/09 (129), 05/25 (145), 06/10 (161), 06/26 (177), 07/12 (193), 07/28 (209), 08/13 (225), 08/29 (241), 09/14 (257), 09/30 (273), 10/16 (289), 11/01 (305), 11/17 (321), 12/03 (337), 12/19 (353) | From 2008 to 2017 |

* Starting from that date, one day was subtracted for each initial date during leap years.

The developed model encompassed ten distinct steps: (1) Mosaics were created for each layer; (2) the sinusoidal projection was transformed to the Universal Transverse Mercator (UTM) system; (3) images underwent clipping; (4) pixel reliability classification was conducted, with the aim of extracting "spurious" pixels (as detailed in Table 2), to ensure data integrity and consistency [59]; (5) image quality was reclassified within the valid range of 4 to 37,572 [60]; (6) the reclassified images were used to generate a "correction mask"; (7) NDVI images were merged with the correction mask, a crucial step to remove pixels affected by clouds or atmospheric interference from earlier stages [61]; (8) conversion of images to floating-point values was executed; (9) images were rescaled to digital values between −1 and +1; (10) images were transformed into vectors, subjected to interpolation using the inverse distance weighted (IDW) method [62], and finally, delimited based on the study area [63].

**Table 2.** Pixel reliability values for the NDVI.

| Pixel Value | Quality | Description | Value after Reclassification |
|---|---|---|---|
| −1 | No data | Unprocessed data | No data |
| 0 | Good data | Can be used with confidence | 0 |
| 1 | Marginal data | * Can be used | 0 |
| 2 | Snow/ice | Target covered by snow or ice | No data |
| 3 | Cloud | Cloud covered target | No data |

* Considers additional quality information. Source: [28], authors' adaptation.

We employed an identical approach for the preprocessing of EVI images, which was executed in a manner akin to that of NDVI images. The data geoprocessing model facilitated the automation and linkage of various stages, necessitating minor modifications to accommodate the processing of all 460 NDVI images, EVI images, their vegetation index quality, and pixel reliability. This adaptation underscores the model's potential versatility for diverse research endeavors.

*2.3. Acquisition and Processing of Meteorological Data*

Meteorological data were sourced from the automatic meteorological stations (AMSs) situated within the Instituto Nacional de Meteorologia (National Institute of Meteorology) (INMET), located in Espírito Santo and its surroundings. Our preference for solely utilizing data from AMSs arises from their capacity to aggregate minute-to-minute observed meteorological values, ensuring automatic availability on an hourly basis [64]. Initially, we selected twenty-one stations, with particular emphasis on those offering a minimum of ten years' worth of meteorological data. The chosen stations included Alegre, Alfredo Chaves, Vitória, Santa Teresa, Linhares, and São Mateus within the boundaries of Espírito Santo, alongside Aimorés and Mantena, municipalities located within the state of Minas Gerais (Figure 2 (label 01)).

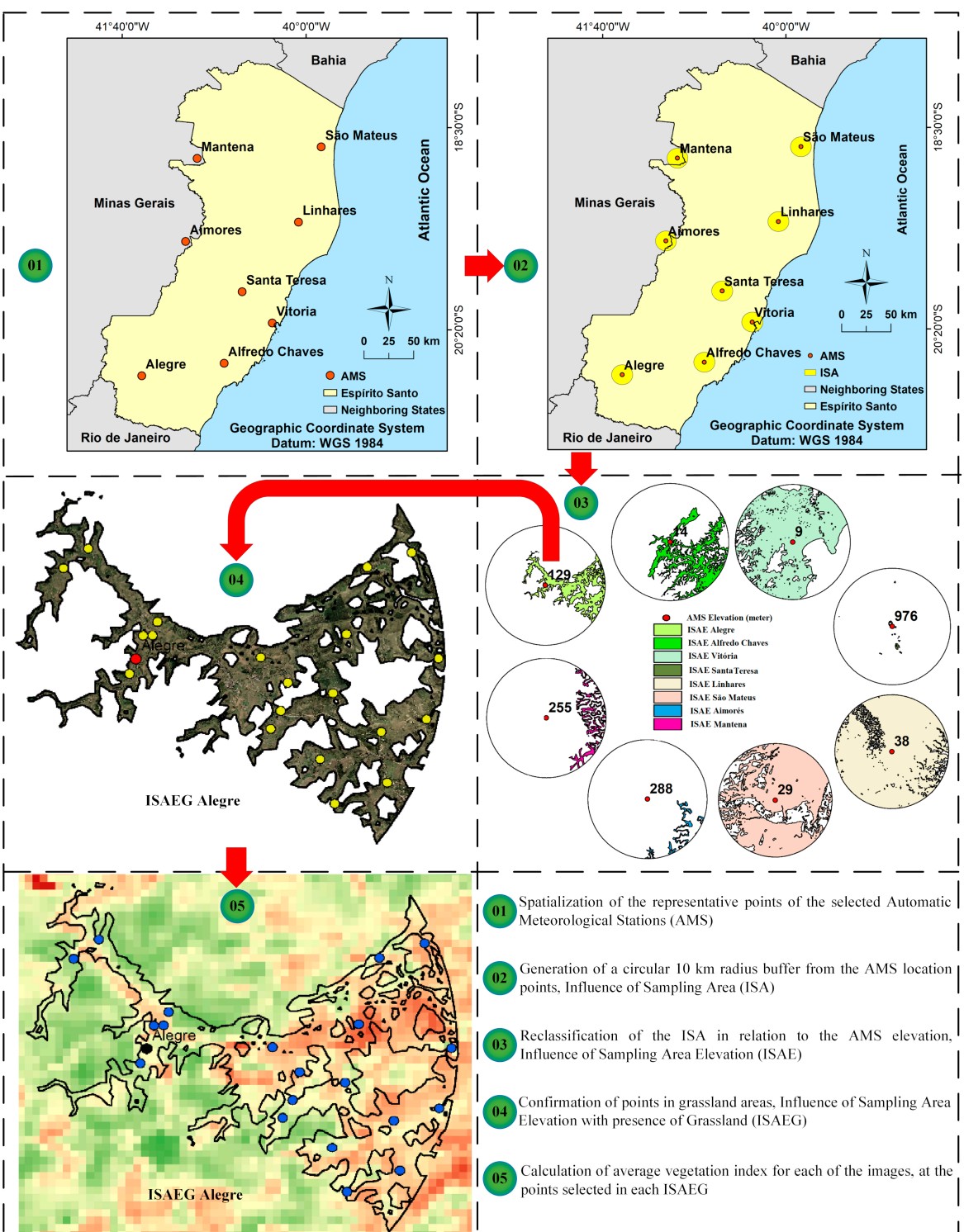

**Figure 2.** Flowchart of a schematic representation detailing the sequential progression of methodologies employed in the process of grassland area selection and subsequent computation of the average vegetation index value.

We conducted an analysis of six meteorological variables, comprising three associated with energy conditions (air temperature, net solar radiation, and reference evapotranspiration) and three pertaining to hydrological conditions (precipitation, relative humidity, and water deficiency). Notably, the reference evapotranspiration was computed using the Penman–Monteith method, a procedure outlined in the Food and Agriculture Organization

of the United Nations' bulletin 56 [65]. Additionally, alongside the daily records for precipitation and evapotranspiration, we computed the daily sequential water balance according to established methods [66] and determined water deficiency using 100 mm of available water capacity as a reference point [67,68].

In relation to the missing data within the variables obtained from AMSs (constituting a fraction of less than 4% of the overall dataset), we opted to utilize the MissForest R package software (ArcGis 10.3) for the purpose of imputation [69]. The selection of this software was driven by its relevance in addressing nonparametric imputation requirements concerning missing values encountered within a diverse spectrum of multivariate datasets. At the core of this approach lies the utilization of a random forest model, which is trained on the observed values within a designated data matrix. This enables the prediction and subsequent completion of the absent values. This technique is well suited for accommodating intricate interactions and nonlinear relationships inherent in the dataset. It is pertinent to note that the MissForest software eliminates the need for calibration via tuning parameters and circumvents any reliance on assumptions regarding the underlying data distribution. An additional noteworthy aspect is its capacity to offer an assessment of input quality. This is realized through the generation of error estimates, which provide a qualitative, albeit averaged, gauge of reliability verification for each variable.

Particularly in the computation of average values for temperature, relative air humidity, and net solar radiation, our focus rested upon utilizing daily data, adhering to the temporal resolution of the vegetation indices' images, which is set at 16 days. Conversely, for rainfall, reference evapotranspiration, and water deficiency, our approach involved aggregating their values over the same time interval. This yielded a total of 230 values for each variable per AMS.

The current study's aim entailed implementing a temporal delay to gauge the interval within which vegetation responds to the effects of meteorological variables. In essence, the data spanning from 13 October 2007 to 31 December 2017 were compiled (Table 3) while incorporating five levels of delay (16, 32, 48, 64, and 80 days) with respect to the dates corresponding to the vegetation index images.

**Table 3.** Different levels of temporal delay for assessing vegetation response to meteorological variables.

| Days of Delay in Relation to Vegetation Index | NDVI or EVI i-th Value ($y_i$) in Response to the i-th Value of the Meteorological Variable ($x_i$) |
|:---:|:---:|
| 0 | $y_i$, with $x_i$ without temporal delay |
| 16 | $y_i$, with $x_i$ previous 16 days |
| 32 | $y_i$, with $x_i$ previous 32 days |
| 48 | $y_i$, with $x_i$ previous 48 days |
| 64 | $y_i$, with $x_i$ previous 64 days |
| 80 | $y_i$, with $x_i$ previous 80 days |

*2.4. Selection of Agricultural Areas Influenced by Meteorological Variables*

On one hand, the ideal approach for assessing the distinct interrelations among the six meteorological variables under investigation, alongside their effects on the NDVI and EVI, would entail the deployment of isolated experimental plots within each AMS. On the other hand, due to substantial requisites of human and financial resources, we were able only to partially explore aspects pertaining to the demarcation of study areas characterized by a singular crop type.

In order to ensure a standardized comparison of the outcomes of vegetation indices in relation to their responses to the effects of meteorological variables, our focus was directed towards a specific crop type possessing the following attributes: (1) exhibiting a relatively swift responsiveness to variations in meteorological conditions; (2) showcasing the highest attainable degree of homogeneity in soil coverage; (3) manifesting minimal susceptibility to topographic shading effects; (4) experiencing limited anthropic changes throughout its

growth cycle; and (5) demonstrating a substantial prevalence within the state of Espírito Santo. Accordingly, we elected to investigate grassland areas as our preferred crop choice, given their alignment with the requisite criteria and their potential to best align with the aims of our research endeavor.

As evident in numerous Brazilian states, grassland areas within Espírito Santo predominantly consist of the *Brachiaria/Urochloa* genus. This species is recognized for its limited nutritional content, inadequate management practices, and lack of appropriate fertilization or soil correction, as documented in previous research [70]. The aforementioned forage species exhibits heightened carbon fixation efficiency within temperature ranges of 30 °C to 40 °C and is categorized as a $C_4$ plant. This classification delineates its photosynthetic mechanism, where $C_3$ plants denote the majority of green plants that produce a three-carbon molecule (3-phosphoglyceric acid) as the initial stable product in the photosynthetic biochemical sequence. Conversely, $C_4$ plants refer to those capable of exceptional dry matter production, adept at fixing $CO_2$ in four-carbon compounds like malate, oxaloacetate, and aspartate [71].

Attention should also be paid to the spatial proximity of the study areas in relation to the AMSs, as we aimed at mitigating potential variance in the meteorological parameter values attributed to geographic distance. In pursuit of this objective, circular zones with radii of up to 10 km were delineated around the precise coordinates of the monitoring stations, facilitating the identification of geographically indicative regions within Espírito Santo. Additionally, areas characterized by minimal variations in elevation were included in the assessment, given the substantial impact of elevation on temperature patterns. Under the premise that elevated altitudes are associated with lower temperatures, as documented in prior investigations [72,73], a temperature gradient of approximately 6.5 °C per one thousand meters alteration in elevation is observed [73,74].

In light of the aforementioned considerations, we executed the subsequent procedures, as follows. (1) Spatialization of the designated AMS points was undertaken (Figure 2 (label 01)). (2) A circular buffer with a radius of 10km was established around each AMS location point, aiming to select areas exhibiting similar meteorological conditions and prevalent grassland occurrences. This designated region was denoted "influence of sampling area (ISA)" (Figure 2 (label 02)). (3) Acquisition of imagery from the digital elevation model, available at http://www.dsr.inpe.br/topodata (accessed on 18 May 2022) [75]. Subsequently to this, the ISA regions were reclassified based on the AMS elevation ±25 m interval. This step was executed to ensure the selection of areas marked by analogous climatic conditions and was labeled "influence of sampling area elevation (ISAE)" (Figure 2 (label 03)). (4) A randomized generation of 100 points was carried out within each ISAE zone. Employing a photointerpretation process, 20 points were singled out from areas characterized by persistent grassland presence over a decade-long span. These selected points were denoted "influence of sampling area elevation with presence of grassland (ISAEG)," with particular emphasis on the ISAEG instances within the municipality of Alegre (Figure 2 (label 04)). Observations: (1) The validation of points situated within grassland areas was conducted utilizing the subsequent resources: (a) a 2007–2008 orthophotomosaic characterized by a 1 m resolution, generously provided by the Instituto Estadual de Meio Ambiente e Recursos Hídricos (IEMA) (State Institute of Environment and Water Resources); (b) a 2012–2015 orthophotomosaic distinguished by a 20 cm spatial resolution; (c) images obtained through the employment of the Web Maps Service (WMS) protocol, encompassing the temporal span of 2008 and 2017. (2) Due to the spatial resolution of the vegetation index images (250 m × 250 m = 6.25 hectares), our selection criteria were oriented towards grassland areas exceeding 10 hectares; (3) the computation of the average NDVI and EVI values was executed for each image, employing a temporal resolution of 16 days. This calculation was performed across the 20 designated points within each ISAEG, spanning the interval from 2008 to 2017 (Figure 2 (label 05)).

*2.5. Statistical Analysis of the Relationship between Meteorological Variables and Vegetation Indices*

Statistical analyses were conducted employing functions within both the R (version 1.5) [76] and GENES [77] software packages (ArcGis 10.3). The identification of outliers within the dataset encompassing all variables under investigation was accomplished by means of box plots. Preceding the analytical procedures, data points considered as outliers were subsequently eliminated from the dataset.

In the course of this investigation, the statistical analysis was executed through a two-step process, as follows.

(1) The calculation of Pearson's coefficients of linear correlation ($r_{xy}$) (Equation (1)) was undertaken, which serves to quantify both the strength and direction of the linear association between a pair of variables [13,78]. This step aimed to quantify the extent of the association between the vegetation indices (specifically NDVI or EVI) and the diverse meteorological variables encompassing (a) air temperature, (b) relative humidity, (c) net solar radiation (defined as the disparity between net shortwave radiation and net longwave radiation, representing the total energy available to influence climatic conditions) [65], (d) precipitation, (e) reference evapotranspiration, and (f) water deficiency. To scrutinize the correlation between the x and y variables, we generated hypotheses formulated as: $H_0$: $\rho = 0$ (indicating statistical equivalence of correlation to zero) and $H_1$: $\rho \neq 0$. The null hypothesis ($H_0$) would be refuted in cases where the computed level (*p*-value), representing the probability of observing a correlation as extreme as the one obtained, is less than or equal to a predetermined significance level ($\alpha$). Here, the significance level ($\alpha$) was preset to 0.01 [79], constituting the threshold beyond which the null hypothesis ($H_0$) would be rejected.

(2) Path analysis involves deconstructing correlations into direct and indirect effects exerted by the explanatory (independent) variables on the basic (dependent) one. These effects are quantified through regression analysis. In essence, path analysis delineates the extent to which each independent variable effects a direct or indirect impact on the dependent variable. Consequently, we conducted path analysis by unpacking the estimated Pearson correlation coefficients ($r_{xy}$) into explicit direct and indirect effects of individual meteorological variables on vegetation indices. The path analysis proceeded through two distinct phases: (a) construction of the path diagram to establish causal relationships among the variables, and (b) decomposition of the correlations into a series of path coefficients. Within this framework, the vegetation indices (NDVI and EVI) were treated as fundamental variables in the model, upon which we examined the direct and indirect effects stemming from six meteorological variables (as explanatory factors).

With application of the aforementioned approach [38], the path coefficients are derivable through Equation (2) and estimations of these path coefficients ($\hat{p}_{0i}$) can be acquired utilizing the least squares method via matrix resolution (Equation (3)). These path coefficients facilitate, for instance, calculations of the direct effects of variable $x_1$ on $y$ within the deconstruction of correlation, denoted as $p_{01}$. Similarly, the indirect effects of variable $x_1$ on $y$ through intermediaries $x_2$ e $x_3$ are represented as $p_{02}r_{12}$ and $p_{03}r_{13}$, respectively. Consequently, $r_{01} = p_{01} + p_{02}r_{12} + p_{03}r_{13} + p_{04}r_{14} + p_{05}r_{15} + p_{06}r_{16}$.

$$r_{xy} = \frac{\sum(x_i - \bar{x}_i)(y_i - \bar{y}_i)}{\sqrt{\left(\sum(x_i - \bar{x}_i)^2\right)\left(\sum(y_i - \bar{y}_i)^2\right)}} \tag{1}$$

Here, $r_{xy}$ represents the Pearson coefficients of linear correlation, with $x$ and $y$ denoting the values of the aforementioned variables.

$$y = p_{01}x_1 + p_{02}x_2 + p_{03}x_3 + p_{04}x_4 + p_{05}x_5 + p_{06}x_6 + p_\varepsilon u \tag{2}$$

In this context, $y$ corresponds to the standardized dependent basic variable (NDVI or EVI), $p_{0i}$ represents the path coefficients or direct effects originated from the meteorological

variables, $x_i$ denotes the explanatory meteorological variables ($i = 1, \ldots, 6$), $u$ embodies the residual variable, and $p_\varepsilon$ stands as the coefficient associated with the residual variable.

$$
\begin{bmatrix} r_{01} \\ r_{02} \\ r_{03} \\ r_{04} \\ r_{05} \\ r_{06} \end{bmatrix} = \begin{bmatrix} 1 & r_{12} & r_{13} & r_{14} & r_{15} & r_{16} \\ r_{21} & 1 & r_{23} & r_{24} & r_{25} & r_{26} \\ r_{31} & r_{32} & 1 & r_{34} & r_{35} & r_{36} \\ r_{41} & r_{42} & r_{43} & 1 & r_{45} & r_{46} \\ r_{51} & r_{52} & r_{53} & r_{54} & 1 & r_{56} \\ r_{61} & r_{62} & r_{63} & r_{64} & r_{65} & 1 \end{bmatrix} \cdot \begin{bmatrix} p_{01} \\ p_{02} \\ p_{03} \\ p_{04} \\ p_{05} \\ p_{06} \end{bmatrix}
\tag{3}
$$

Before delving into the examination of correlation and path coefficients, as depicted in Table 4, it was imperative to verify the presence of multicollinearity, in order to avoid any potential misinterpretations in the outcomes, as highlighted in certain scholarly investigations [80]. To accomplish this verification, we concurrently employed the following methodologies. (a) number of conditions (NCs), which is computed through the ratio between the largest and smallest eigenvalues of the matrix. If the NCs is below 100, it signifies weak multicollinearity with minimal effect on the analysis. If the NCs lies between 100 and 1000, it denotes a moderate to strong multicollinearity presence. A value exceeding 1000 indicates severe multicollinearity [81]. (b) Variance inflation factor (VIF), which gauges the extent to which the variance of a coefficient is elevated in comparison to what it would be if the variable were not correlated with any other. Therefore, all VIF values should be below 10, indicating an absence of multicollinearity influence [82].

**Table 4.** Criteria for the interpretation of path analysis.

| Condition | Interpretation |
|---|---|
| 1st—if both $r_{xy}$ and path coefficient were statistically significant in magnitude and signal | Direct effect of the explanatory variable |
| 2nd—if $r_{xy}$ exhibited significance, whereas the path coefficient did not achieve statistical significance | Correlation arose due to indirect effects |
| 3rd—if $r_{xy}$ lacked significance, whereas the path coefficient demonstrated statistical significance | Existence of a direct effect of the variable, however the absence of correlation was attributed to the presence of indirect effects |
| 4th—if neither $r_{xy}$ nor the path coefficient reached statistical significance | Effects that did not achieve statistical significance (ns) |

Source: Adapted from [80].

Subsequently, two approaches were investigated to mitigate the effects of variables influenced by multicollinearity: (a) the exclusion of highly correlated explanatory variables and (b) the application of ridge regression, a technique utilized to estimate path coefficients while determining the optimal course of action to rectify the influence of multicollinearity. This method has demonstrated favorable outcomes in various research endeavors [83,84]. For variables unaffected by multicollinearity, standard analytical procedures were employed.

## 3. Results and Discussion

### 3.1. Preliminary Correlation Analysis between Meteorological Variables and Vegetation Indices

Research concerning the intricate dynamics of vegetation growth and variations has consistently yielded analogous findings when employing either the normalized difference vegetation index (NDVI) or the enhanced vegetation index (EVI). Nonetheless, some scholars state that the performance of the EVI surpasses that of the NDVI, primarily attributed to enhancements in its capacity for vegetation monitoring. These improvements encompass the segregation of signals originating from the lower canopy, attenuation of interferences stemming from soil and atmospheric affects, and a decelerated saturation tendency in areas characterized by elevated vegetation density [35,85–87].

The primary objective of this research endeavor is to enhance our comprehension of the intricate interplay between meteorological factors and vegetation indices. In pursuit of this

aim, a detailed investigation was undertaken to delineate the correlations existing between vegetation indices and individual meteorological variables. The analysis encompassed varying time lags, specifically applied in the geographic areas of Alegre, Aimorés, and Mantena, which were chosen through a randomized selection process from a pool of eight AMSs (Figure 3).

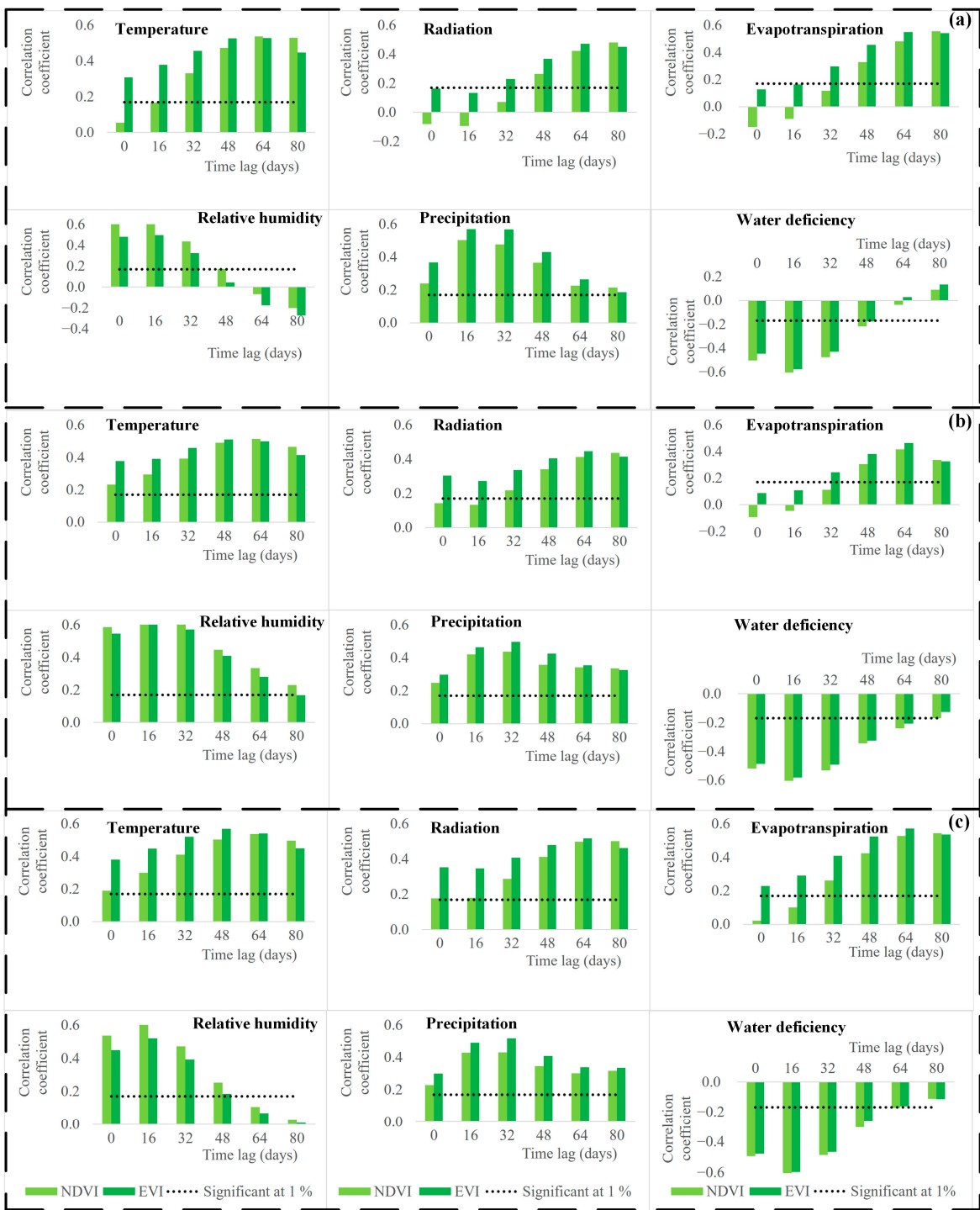

**Figure 3.** Correlation coefficients between vegetation indices and individual meteorological variables across the localities of Alegre (**a**), Aimorés (**b**), and Mantena (**c**), considering five distinct levels of temporal delay (days).

Both the NDVI and EVI exhibited congruent patterns of response to the diverse meteorological variables across the distinct geographic areas under scrutiny. Notably, this correspondence held true for all three distinct areas, characterized by different temporal delay periods encompassing five discrete levels. Remarkably, the EVI demonstrated superior correlation coefficients to the NDVI in the majority of associations, barring the case of the variable pertaining to relative humidity.

As depicted in Figure 3, the absolute values of correlation coefficients lack strength in any association between meteorological variables and vegetation indices [88]. Nevertheless, it is noteworthy that the EVI has exhibited a higher incidence of statistically significant correlations in comparison to the NDVI across the three delineated areas. This observation substantiates findings from a previous investigation conducted at the Reserva Biológica de Sooretama (Sooretama Biological Reserve) situated within Espírito Santo. The aforementioned research in the same location similarly underscored EVI's superior performance vis-à-vis NDVI correlations with identical meteorological variables [89].

We draw the reader's attention to noteworthy variables demonstrating either positive or negative correlations with vegetation indices. When considering the most robust correlation coefficients between individual meteorological variables and vegetation indices, it is pertinent to highlight that temperature, relative air humidity, net solar radiation, precipitation, and reference evapotranspiration exhibited positive correlations. This suggests that an increase in these variables is accompanied by a concomitant rise in vegetation indices, while a decrease in these variables corresponds to a decline in indices. On the other hand, water deficiency emerged as the sole variable displaying a negative correlation with vegetation indices—a phenomenon that is normally expected. As water deficiency intensifies, there is a corresponding decrease in plant growth, underscoring the inverse relationship. Conversely, augmented vegetation growth is associated with a surplus of available water.

In relation to the NDVI, a study underscored its positive correlation with average temperature, precipitation, and relative humidity. Notably, the strongest correlation was observed with average temperature, yielding a correlation coefficient of 0.7874 [90]. Further insights into NDVI dynamics can be gleaned from an investigation that scrutinized the variations in vegetation cover and its correlation with climatic factors in Mongolia spanning the period 1982 to 1999. The outcomes of this study revealed that the average vegetation cover during the growth season exhibited a positive correlation with precipitation, with an average correlation coefficient of 0.6460 [91].

Scholars additionally delved into the investigation of the interrelationship between the NDVI and the leaf area index (LAI) concerning climatic parameters, encompassing maximum temperature, minimum temperature, relative humidity, precipitation, wind speed, and aerosol optical depth, within the geographic context of Haryana, India, spanning the temporal scope of 2010 to 2020. The correlation coefficient values of both the NDVI and LAI with the aforementioned climatic variables exhibited temporal variations across months, although the fundamental essence of their variations remained similar for the two indices. The overall correlation analysis unveiled a positive correlation between rainfall and relative humidity with both the NDVI and LAI. In contrast, the remaining climatic variables exhibited a negative correlation in relation to the indices [92].

Several scholars have investigated the interannual variability in vegetation coverage and its responses to meteorological variables, along with the effects of climatic changes on the grassland ecosystem within the northern Tibetan region of China. This inquiry was underpinned by an analysis of an NDVI dataset and meteorological records spanning from 2000 to 2009. The findings of this study unveiled substantial positive linear associations between the NDVI values and hydrological as well as thermal conditions. Among these associations, the connection between the monthly mean minimum temperature and the NDVI exhibited the most robust correlation. Furthermore, a delayed response of the NDVI to alterations in hydrological and thermal conditions was identified. The temporal lag observed in this response was approximately one month for both precipitation and

temperature variables. However, in the case of cumulative precipitation, the lag extended to two months [93].

Several scholarly investigations have highlighted the significance of net solar radiation not only as the principal energy source driving plant biomass production but also as a potent factor exhibiting a robust positive correlation with reference evapotranspiration. It has been posited that these variables are intimately interlinked within the soil–plant–atmosphere system, profoundly affecting water movement and exerting considerable influence on various vegetation indices [94].

Within the context of three distinct geographic areas (Figure 3), it was discerned that water deficiency exhibited the most substantial absolute correlation coefficients when juxtaposed with vegetation indices, particularly when assessed with a sixteen-day temporal offset. This observation potentially signifies a comparatively faster responsiveness of vegetation to instances of water scarcity or excess. However, the validation of this proposition necessitates further substantiation through path analysis methodology.

Temperature and precipitation emerged as meteorological variables of paramount significance, exhibiting substantial correlations with both vegetation indices across a range of temporal delay levels. This observation aligns with their recurrent utilization in scholarly inquiries concerning plant dynamics and developmental patterns [15,17]. Nonetheless, the extent of their influence on vegetation indices, whether through direct causation or indirect associations with other variables, was examined using a path analysis framework.

In this context, pertinent studies propose that temperature escalation, within a certain limit, holds the potential to accelerate vital metabolic processes linked to plant growth. For instance, heightened temperatures can stimulate photosynthesis, thereby fostering an increase in vegetation indices. Nevertheless, it is imperative to acknowledge that an excessive increase in atmospheric temperature could potentially intensify vegetation transpiration rates, ultimately precipitating a scarcity of available water for vegetation growth. This becomes particularly evident in areas characterized by water deficits [95].

Notably, other scholarly investigations state that precipitation emerges as a prominent determinant influencing the course of plant development. During phases characterized by increased precipitation, there is a distinct vigor in vegetation growth. Conversely, instances of diminished rainfall are accompanied by a contraction in growth rates, leading to substandard vegetation development [96,97].

Of particular relevance, the majority of grasses constituting the grassland ecosystem ($C_4$ plants) within the study area exhibit a marked affinity for $CO_2$. This characteristic engenders a concomitant elevation in net photosynthetic rates, coupled with a heightened light saturation point [71]. Therefore, during periods marked by elevated solar radiation and elevated temperatures, these $C_4$ plants possess the capacity to generate a greater biomass output compared to their $C_3$ and crassulacean acid metabolism (CAM) plant counterparts. This heightened biomass production contributes to an overall increase in the vegetation index.

Time lags resulting from the delay in vegetation's responsiveness to meteorological variations hold significance equivalent to the associations between vegetation indices and meteorological variables. Delving into the time intervals that encompass this delay assumes pivotal importance across domains concerning planting, harvesting, management strategies, and safeguarding of ecosystems [17]. Illustrated in Figure 3, the response of vegetation to meteorological variables pertaining to energy dynamics (such as temperature, net solar radiation, and evapotranspiration, depicted in the first lines of graphs (a), (b), and (c)), can exhibit a relatively extended timeframe (ranging from 48 to 80 days). Conversely, responses attributed to hydrological variables (including relative humidity, precipitation, and water deficiency, represented in the second lines of graphs (a), (b), and (c)) tend to manifest at a comparatively faster pace (spanning 0 to 32 days). In order to facilitate a more comprehensive and appropriate investigation into the underlying causal relationships between meteorological variables and vegetation indices, we opted to utilize the path analysis methodology.

*3.2. Path Analysis of Meteorological Variables and Vegetation Indices*

It is essential to underscore that meteorological variables can also exhibit intercorrelations, giving rise to issues of multicollinearity in the context of path analysis. In pursuit of obtaining path coefficients characterized by reliable accuracy, the presence of multicollinearity was initially identified and subsequently mitigated. In the case of the area of Aimorés, the path analysis unveiled number of conditions (NCs) values below 100 [81], along with variance inflation factor (VIF) values below 10 across all five temporal delays [82]. Consequently, the analysis was conducted without necessitating adjustments for multicollinearity-induced biases. Conversely, within the domains of Alegre and Mantena, the degree of multicollinearity exhibited a scale ranging from moderate to strong. NC values exceeded 100 for all five temporal delays, and VIF values surpassed 10 specifically for net solar radiation and reference evapotranspiration variables. In light of this, prior to initiating the analysis, requisite measures were undertaken to rectify the potential influence of multicollinearity effects.

In addressing the prevailing concern, our initial strategy involved the deliberate exclusion of the reference evapotranspiration variable. This variable exhibited a noteworthy correlation coefficient with net solar radiation and temperature. By adopting this course of action, a substantial improvement in NC values, well below the threshold of 100, was achieved across the five remaining variables. Similarly, a reduction in VIF values, falling below 10, was accomplished. However, it is important to acknowledge that the exclusion of variables may involuntarily lead to the loss of crucial information concerning the causal relationship between the eliminated variable and vegetation indices. Given this particular situation, we employed the ridge regression method, which presents the distinct advantage of not necessitating the exclusion of variables in the analytical process [81]. To delineate the procedure briefly, the parameter values for the constant *k* were carefully identified through the visual examination of the crest trace [98]. This attempt involved the selection of the lowest *k* value capable of effectively enhancing the stability of a majority of path coefficient estimators. Subsequently, this selected *k* value was integrated into the main diagonal of the correlation matrix. This strategic manipulation served to effectively manage the multicollinearity effects present within the dataset. The consequential outcome of this particular process was the attainment of NC and VIF values that adhered to appropriate and suitable intervals, thus establishing the methodological reliability requisite for our analytical pursuits.

A causal diagram was constructed to facilitate path analysis. In this diagram, the basic dependent variable was represented by either vegetation index, specifically the NDVI or EVI. The independent explanatory variables comprised temperature, relative humidity, net solar radiation, precipitation, reference evapotranspiration, and water deficiency. Additionally, a residual variable ($\varepsilon$) was included in the model to account for unexplained variance. This diagram encapsulated the intricate relationships among these variables, forming a foundation for rigorous causal inference and analytical exploration within this domain, as evidenced by prior contributions in the field.

Drawing upon an adaptation of the proposed method, we embarked on the interpretation of coefficients to discern the direct and indirect effects [80]. By employing this approach, we were able to elucidate the intricate relationships between variables. To illustrate, we present a pattern of the outcomes from the path analysis in Table 5. The table showcases coefficient values, encompassing both direct and indirect effects, specifically in the context of EVI analysis within the Aimorés area. This particular presentation extends to scenarios devoid of any temporal delay, featuring corresponding values for the Pearson correlation coefficient ($r_{xEVI}$), coupled with an assessment of multicollinearity diagnostics.

**Table 5.** Direct effects or path coefficients (main diagonal) and indirect effects (other values) in the EVI context, excluding temporal delays, for Aimorés, Minas Gerais; Pearson correlation coefficient ($r_{xEVI}$) and multicollinearity diagnosis.

| VARIABLES ($x_i$) | Temperature | Relative Humidity | Solar Radiation | Precipitation | Evapotranspiration | Water Deficiency |
|---|---|---|---|---|---|---|
| Temperature | **0.336** | −0.102 | 0.533 | −0.024 | −0.322 | −0.041 |
| Relative humidity | −0.059 | **0.580** | −0.220 | −0.145 | 0.192 | 0.199 |
| Solar radiation | 0.267 | −0.191 | **0.671** | 0.007 | −0.372 | −0.078 |
| Precipitation | 0.034 | 0.355 | −0.020 | **−0.237** | 0.018 | 0.148 |
| Evapotranspiration | 0.263 | −0.271 | 0.606 | 0.010 | **−0.412** | −0.110 |
| Water deficiency | 0.057 | −0.473 | 0.215 | 0.140 | −0.186 | **−0.244** |
| $r_{xEVI}$ | **0.378** | **0.547** | **0.304** | **0.298** | **0.086** | **−0.487** |
| Multicollinearity diagnosis | | | | | | |
| Number of Conditions | | | 45.53 | | | |
| Variance inflation factor | 3.17 | 3.68 | 6.56 | 2.11 | 8.72 | 3.43 |

Upon analyzing the highlighted values along the main diagonal, in conjunction with the corresponding correlation coefficients between meteorological variables and the EVI ($r_{xEVI}$), it was determined that temperature, relative humidity, net solar radiation, and water deficiency exhibited direct effects on the EVI. This determination was supported by the presence of noteworthy correlation values of $r_{xEVI}$ (both in magnitude and signal) as well as notable path coefficients. In the case of precipitation, while $r_{xEVI}$ correlation exhibited statistical significance and the corresponding path coefficient did not, suggesting that this correlation resulted from indirect effects. These effects were predominantly attributed to the interrelation between precipitation and relative humidity, as well as water deficiency. In contrast, reference evapotranspiration demonstrated no discernible effect on the EVI under these conditions, as neither $r_{xEVI}$ nor the path coefficient yielded significance.

In light of the aforementioned observations, the causal diagram (Figure 4) was adapted from the GENES software [77]. This diagram visually elucidates the path coefficients, representing direct effects of meteorological variables (namely, temperature, relative humidity, net solar radiation, precipitation, reference evapotranspiration, and water deficiency) on the EVI through single arrows. Additionally, it illustrates the correlations existing between the explanatory or independent variables via double arrows.

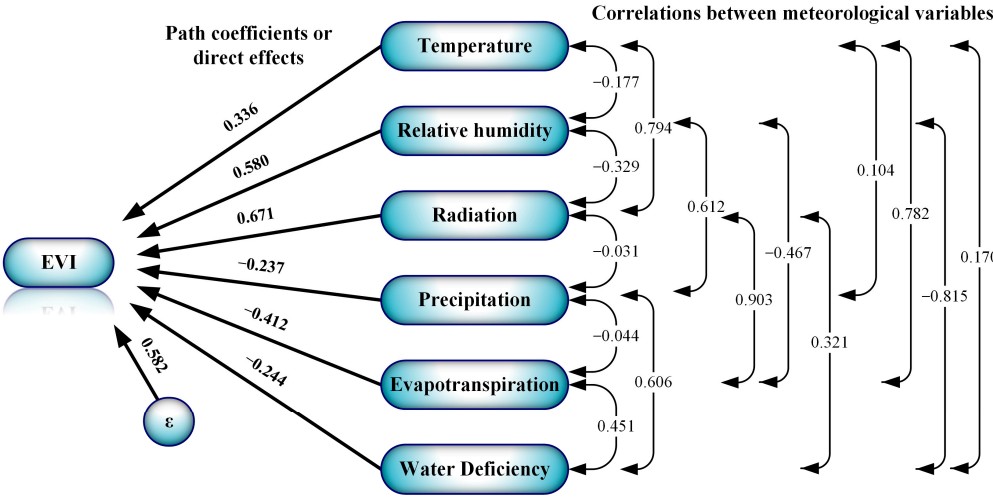

**Figure 4.** Causal diagram depicting the path coefficients of meteorological variables influencing EVI (indicated by single arrows), as well as the correlations existing among these variables (denoted by double arrows) for Aimorés, Minas Gerais, with no inclusion of temporal delay.

An identical analysis was executed for each of the remaining study areas, as well as for the various temporal delays under consideration in relation to the vegetation response to meteorological variables. Figure 5 provides a comprehensive overview encompassing all six meteorological variables, elucidating both the direct and indirect effects on the NDVI and EVI across five distinct delay levels for the localities of Alegre (a), Aimorés (b), and Mantena (c). Upon examination, the dynamic nature of variations in vegetation indices and meteorological variables became evident. Nevertheless, it is of significance to highlight that the direct and indirect effects of meteorological variables on the NDVI and EVI exhibited striking similarities across the three areas, thus reiterating a strong correlation between these two indices.

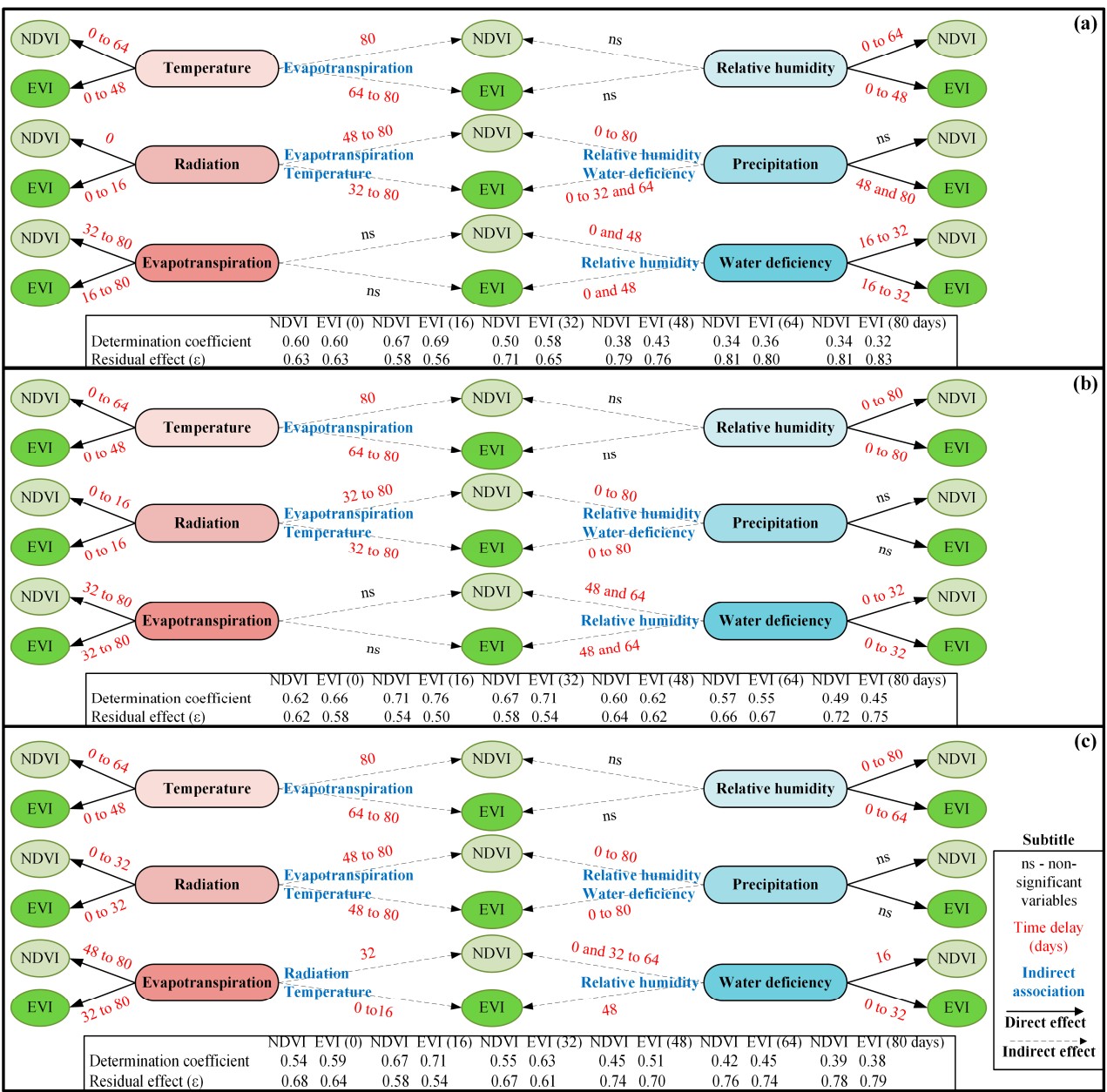

**Figure 5.** Direct and indirect effects of meteorological variables on vegetation indices within the locations of Alegre (**a**), Aimorés (**b**), and Mantena (**c**), which were examined across five distinct levels of temporal delay (days).

The determination coefficients ($R^2$) indicate that the set of six meteorological variables explained over 50% of the variation observed in both the NDVI and EVI, with potential

explanatory power extending to around 70% in certain instances, particularly during the initial delay periods characterized by a slight elevation in the EVI over the NDVI. However, the substantial residual effect ($\varepsilon$) indicates the potential influence of unaccounted variables in the path analysis, which might have impacted the vegetation indices, particularly during the latter delay periods.

As previously indicated, the study area of Aimorés did not require the application of multicollinearity correction, thereby yielding higher $R^2$ values. On the other hand, the areas of Alegre and Mantena required corrections through the utilization of a ridge regression-based approach, leading to a diminution in $R^2$ and an elevation in residual components when contrasted with the uncorrected analysis. However, it is noteworthy that both $R^2$ and the residual effect exhibited congruent patterns of variation akin to those observed in the Aimorés region.

Another aspect to be highlighted is that in cases where variables related to energy (temperature, net solar radiation, and reference evapotranspiration) or hydrology (relative humidity, precipitation, and water deficiency) exhibited indirect effects on the NDVI and EVI. Such effects arose through associations with variables that fall within the same group, i.e., hydrological variables exhibited correlations among themselves, whereas energy variables conformed to the same underlying process.

Relative humidity did not exhibit indirect effects on vegetation indices. Nonetheless, this observation does not infer the variable's complete independence, as other variables demonstrated associations with relative humidity through indirect effects. By contrast, the impact of precipitation was primarily channeled through indirect effects impacting the EVI and NDVI, mediated by relative humidity and water deficiency. These findings underscore the notion that precipitation seldom operates in isolation, but rather is nearly invariably intertwined with other hydrological variables when exerting its influence on vegetation indices.

For the purpose of enhancing the visualization and facilitating comparative analysis of outcomes, Figure 6 provides a distinct presentation of the path coefficients, indicating significant direct effects, stemming from meteorological variables onto the vegetation indices within the three delineated areas (Alegre, Aimorés, and Mantena), while considering various temporal delays.

It is noteworthy that reference evapotranspiration and water deficiency exhibited contrasting dynamics. Evapotranspiration demonstrated a discernible direct influence on both vegetation indices, primarily following a delay of 32 days. On the other hand, the direct effect of water deficiency persisted only up to the 32-day mark, implying that the vegetation displayed a relatively prompt response to variations in water deficiency or abundance.

In a particular study centered on the dynamics of the NDVI within a metropolitan region situated in northern China characterized by a semiarid climate, a comprehensive assessment of changing climatic influences was conducted. Scholars examined three distinct factors—precipitation as a representative of water conditions, net solar radiation, and air temperature—to represent energy conditions [9]. The investigation unveiled substantial variations in NDVI responses contingent upon climatic factors. Broadly, the NDVI response time interval related to energy factors surpassed that of water-related factors. This pattern of behavior aligns notably with findings from analogous investigations conducted in diverse geographic settings [96,99,100]. However, it is imperative to underscore that an analysis of causality should not solely rely on correlation coefficients.

Consequently, exercising caution is recommended when drawing inferences concerning the temporal responsiveness of vegetation in relation to meteorological variables associated with energy or water dynamics. While an exclusive consideration of correlation coefficients aligns our findings with the aforementioned study [9], divergent conclusions emerge when evaluating the path coefficients, which denote direct effects. Figure 6 provides a comprehensive overview of how the temporal response intervals of the NDVI and EVI exhibited variations relative to meteorological variables. The response to net solar radiation

was predominantly evident within the initial three delay periods, spanning up to 32 days. Air temperature and relative humidity emerged as the primary drivers exerting direct influences on vegetation indices across nearly all delay phases. Reference evapotranspiration engendered a discernible response primarily within the latter four delay levels, while water deficiency exhibited a response primarily within the initial three periods of delay.

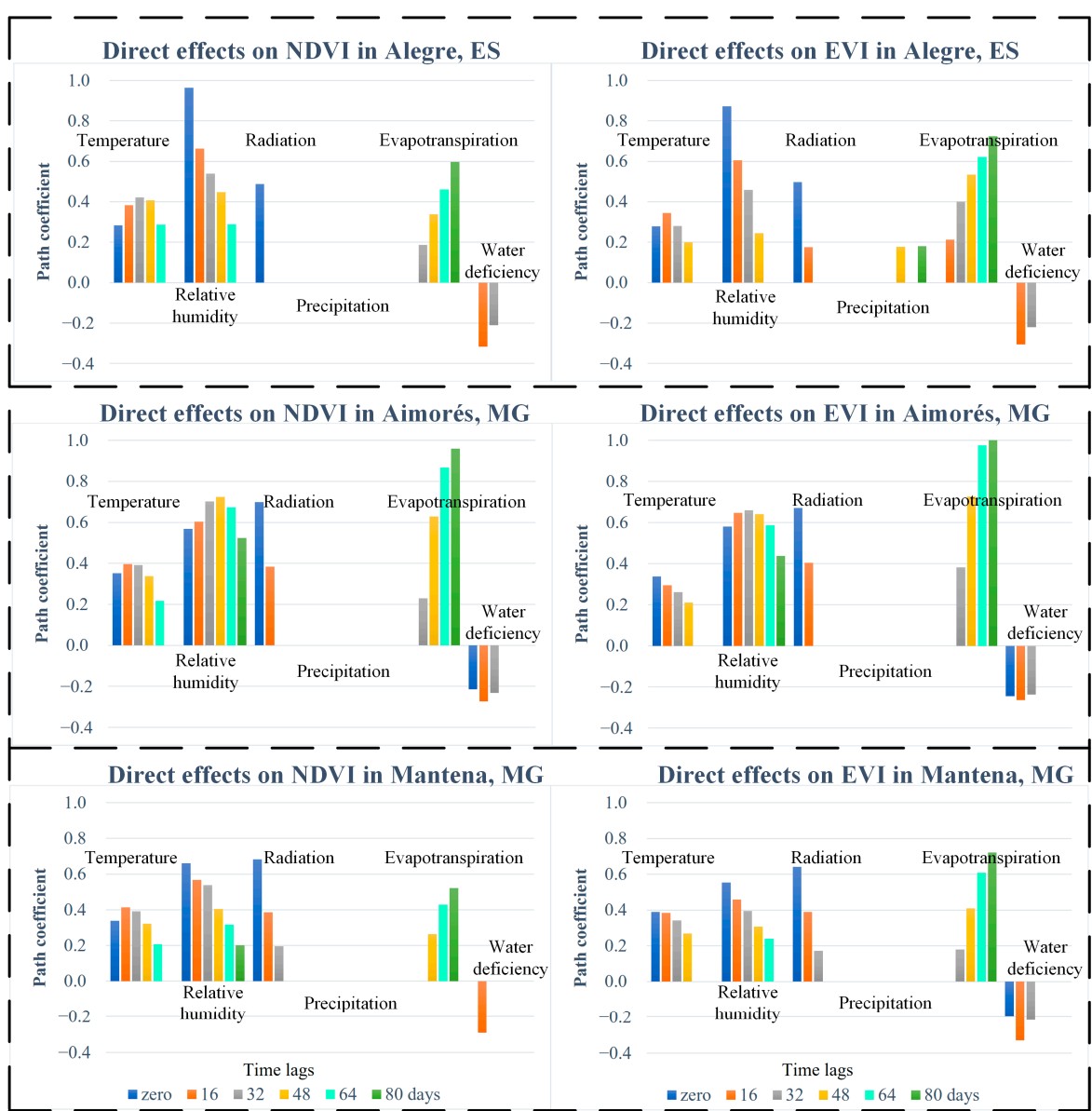

**Figure 6.** Path coefficients denoting significant direct effects originating from meteorological variables onto vegetation indices, examined across five distinct levels of temporal delay.

Precipitation did not exhibit direct effects in either index, although this observation should not be misconstrued as implying a lack of impact on the development of grasses and, by consequence, the composition of grasslands. Rather, it underscores the prevailing notion that precipitation typically operates in concert with other hydrological variables, notably water deficiency and relative humidity. Therefore, precipitation manifests indirect effects on vegetation indices, given that its mere occurrence does not inherently signify available water for plant growth, owing to the presence of associated factors that contribute to this phenomenon.

Two studies conducted in China employing path analysis to assess the effects of precipitation and temperature on vegetation growth based on the NDVI yielded similar results.

Within the semiarid and pastoral region of Ordos City, in the northern part of the country, the authors concluded that precipitation held greater significance as a climatic factor influencing vegetation growth compared to temperature. Enhanced precipitation levels directly promoted vegetation growth, whereas temperature elevation exhibited a damaging influence by diminishing overall precipitation during the entire growing season [55]. In another investigation within the Ziya–Daqing basins, the effects of temperature and precipitation on vegetation growth were predominantly conveyed through direct effects.

Across 42% of the analyzed area where the path analysis demonstrated efficacy and the NDVI exhibited significant correlation with temperature, the magnitude of the direct temperature effect exceeded its indirect impact. In 58% of this territory, the magnitudes of direct and indirect effects were found to be comparable. Within 23% of the basins, interannual NDVI variations were predominantly governed by the direct effect of precipitation, while in 5%, the interplay of precipitation and temperature exerted direct effects. In less than 1% of the cases, the interplay of direct temperature effect or indirect impacts of these climatic factors held dominance. Importantly, the direct temperature effect was largely negative, signifying that an annual rise in temperature detrimentally affected vegetation growth. In contrast, the majority of the direct effect stemming from precipitation was positive, indicating that an increase in annual precipitation was favorable to enhanced vegetation growth [54].

Upon careful examination of the aforementioned elucidations, it becomes evident that comprehending the intricate interconnections between meteorological variables and vegetation dynamics necessitates progressively advanced and complex investigations. Therefore, it is crucial to emphasize that our study contributes to a broad comprehension of the associations among meteorological variables and vegetation indices, specifically the NDVI and EVI, albeit within a restricted set of meteorological variables under tropical climate conditions. Nonetheless, it remains fundamental to encompass the interplay of these or other vegetation indices with the entirety of local meteorological variables to achieve a more nuanced understanding of the dynamics governing plant development across the investigated areas.

## 4. Conclusions

The responses of vegetation to climate factors involve intricate processes. However, given the specific aims of this research, not to mention its methodological underpinnings, we have drawn the following conclusions. (1) The utilization of path analysis facilitated the comprehensive assessment of both direct and indirect effects of meteorological variables on vegetation indices. This analytical approach proves to be highly advantageous, considering that many studies employ correlation coefficients indiscriminately to examine causal relationships among variables. (2) Regarding EVI outcomes, they exhibited superiority over NDVI results in terms of the quantity of significant correlations with meteorological variables, as well as the responses to both direct and indirect effects of variables within grassland areas. (3) The effects of meteorological variables on vegetation indices exhibited variations contingent upon distinct delay periods. Consequently, such analyses become vital for developing strategies pertaining to planting, harvesting, ecosystem preservation, and management. (4) Indirect effects of meteorological variables on both the NDVI and EVI occurred in conjunction with variables belonging to the same group, implying interactions within hydrological variables or between variables linked to energy. (5) Direct and indirect effects of the aforementioned variables on vegetation indices displayed notable consistency across different areas and temporal delays. (6) Notably, our methodological framework retains its adaptability to various regions within Brazil and potentially on a global scale. Such adaptations could be employed to ascertain similarities or disparities in obtained outcomes, involving a broader spectrum of meteorological variables and diverse vegetation indices in future research endeavors.

**Author Contributions:** Conceptualization, A.P.S., E.F.d.S., A.R.d.S., J.L.F. and J.B.E.P.; methodology, A.P.S., E.F.d.S., A.R.d.S., J.L.F., J.B.E.P. and R.d.C.F.C.; software, A.P.S., A.R.d.S., K.B.d.S., T.R.M. and R.d.C.F.C.; validation, A.P.S., E.F.d.S., A.R.d.S., K.B.d.S., T.R.M. and R.d.C.F.C.; formal analysis, A.P.S.; investigation, A.P.S.; resources, A.P.S., E.F.d.S., A.R.d.S. and R.d.C.F.C.; data curation, A.P.S.; writing—original draft preparation, A.P.S.; writing—review and editing, A.P.S., E.F.d.S., A.R.d.S., J.L.F. and J.B.E.P.; visualization, A.P.S., E.F.d.S., A.R.d.S., K.B.d.S., T.R.M. and R.d.C.F.C.; supervision, A.P.S., E.F.d.S. and A.R.d.S.; project administration, A.P.S.; funding acquisition, A.P.S., E.F.d.S. and A.R.d.S. All authors have read and agreed to the published version of the manuscript.

**Funding:** The authors thank the following research and development agencies for their assistance, funding, and support: (a) the Fundação de Amparo à Pesquisa e Inovação do Espírito Santo (FAPES)—FAPES Public Notice n. 05/2023, (b) Coordination for the Improvement of Higher Education Personnnel (CAPES), and (c) the National Council for Scientific and Technological Development (CNPq).

**Institutional Review Board Statement:** Not applicable.

**Data Availability Statement:** The data are contained within the article.

**Acknowledgments:** The authors thank the Federal Institute of Education, Science, and Technology of Espírito Santo, the Postgraduate Program in Plant Production at the State University of North Fluminense Darcy Ribeiro (UENF), the Postgraduate Program in Forest Sciences at the Federal University of Espírito Santo, and last but not least the CNPq research group Geotechnology Applied to the Global Environment (GAGEN). We also thank the National Aeronautics and Space Administration (NASA), the Integrated System of Geospatial Bases of the State of Espírito Santo (GEOBASES), the National Institute of Meteorology (INMET), and the Instituto Estadual de Meio Ambiente e Recursos Hídricos (IEMA) for providing the necessary data for the implementation of this work.

**Conflicts of Interest:** The authors declare no conflict of interest. The funders had no role in the design of the study; in the collection, analyses, or interpretation of data; in the writing of the manuscript; or in the decision to publish the results. This manuscript has not been published or presented elsewhere in part or in its entirety and is not under consideration by another journal.

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
