# Peer review of "Application of Path Analysis and Remote Sensing to Assess the Interrelationships between Meteorological Variables and Vegetation Indices in the State of Espírito Santo, Southeastern Brazil"

_diversity, doi:10.3390/d16020090_

Round 1

Reviewer 2 Report

Summary:

This study used path analysis to evaluate the role of various meteorological variables in driving NDVI and EVI in Espirito Santo, Brazil. I thought that the use of path analysis was quite interesting, however the results were difficult to follow and additional descriptions are needed to help someone who is not a specialist with this technique. There is also a need to improve the grammar in the paper in many places. I think that the paper should undergo major revisions and be reviewed again. I outline my specific questions/comments below.

Comments/Questions:

The grammar of the manuscript needs to be improved/revised throughout.

The title and abstract of the paper make no mention of the limited geographic focus of the analysis in this study. I think that the study location, Espirito Santo, Brazil, should be stated in both places.

L166-167: briefly describe how data were gap filled by this algorithm. What percentage of data were missing?

L246: what is meant by “liquid” solar radiation? Please clarify.

L258-286: A little more background on how this technique works and is applied would be helpful here. It looks like the path coefficients are calculated based on the matrix in equation (3) but it seems like one or two sentences are missing in the text on how that is done. How are the “number of conditions” and “variance inflation factor” parameters calculated?

Figure 3 and 4 seem to be nearly identical but Figure 3 is made up of line graphs while Figure 4 is bar charts. Is there a difference? It doesn’t seem so, so one should be removed if they are redundant.

Are those path coefficients in shown in Table 5? Please clarify in the caption.

Figures 5-6 are quite complex but receive very little description in the text after being cited around L434-449 and are almost impossible to interpret (especially Figure 5) on their own. A bit more text is needed to describe the results displayed in these figures.

Figure 5: The yellow circles/numbers/letters need to be described in the figure caption. Perhaps adjusting the thickness and/or color of the arrow lines might help to highlight the strongest relations/paths.

Figure 6: The caption needs more information to help identify the different components of this large and complex figure. What are the numbers in the tables at the bottom of each panel? Why are some arrows solid while others are dashed lines?

Figure 7: do larger path coefficients mean a greater effect?

Look out for some additional publications that did similar work with path analysis and NDVI including:

Li J. Responses of Vegetation NDVI to Climate Change and Land Use in Ordos City, North China. Applied Sciences. 2022; 12(14):7288. https://doi.org/10.3390/app12147288

Long, S., Qingxi, G., Wenchao, Z., Guoting, Y., & Xiaofeng, Z. (2005). The path analysis on NDVI of typical vegetations and climate factors in North-South Transect of Eastern China. Journal of Northeast Forestry University, 33(5), 59-61.

Shao, W., Wang, Q., Guan, Q., Luo, H., Ma, Y., & Zhang, J. (2022). Distribution of soil available nutrients and their response to environmental factors based on path analysis model in arid and semi-arid area of northwest China. Science of The Total Environment, 827, 154254.

Fatemi, M., & Jebali, A. (2022). Path analysis of the effect of climatic elements on wind speed and desertification progress in Central Iran. Arabian Journal of Geosciences, 15(10), 930.

The grammar of the manuscript needs to be improved/revised throughout.

Round 2

Reviewer 2 Report

The authors did a very thorough job reworking the text and I feel addressed all of my comments and suggestions. In the markup manuscript there appears to be a repeat of Figure 2 on page 8, however that could be corrected in the proof stage.

There are a few typos and/or grammar issues but fine overall.